# A Prospective, Case-Control Study of Serum Metabolomics in Neonates with Late-Onset Sepsis and Necrotizing Enterocolitis

**DOI:** 10.3390/jcm11185270

**Published:** 2022-09-07

**Authors:** Agathi Thomaidou, Olga Deda, Olga Begou, Artemis Lioupi, Angeliki Kontou, Helen Gika, Eleni Agakidou, Georgios Theodoridis, Kosmas Sarafidis

**Affiliations:** 11st Department of Neonatology, School of Medicine, Aristotle University of Thessaloniki, Hippokrateion General Hospital, 54642 Thessaloniki, Greece; 2Laboratory of Forensic Medicine and Toxicology, School of Medicine, Aristotle University of Thessaloniki, 54124 Thessaloniki, Greece; 3Biomic_AUTh, Center for Interdisciplinary Research and Innovation (CIRI-AUTH), Balkan Center, 54124 Thessaloniki, Greece; 4School of Chemistry, Aristotle University of Thessaloniki, 54124 Thessaloniki, Greece

**Keywords:** biomarkers, diagnosis, lipids

## Abstract

Late-onset sepsis (LOS) and necrotizing enterocolitis (NEC) are major causes of neonatal morbidity and mortality. In this prospective, case-control study, we evaluated the metabolic profile of neonates with LOS and NEC. Blood samples were collected from 15 septic neonates and 17 neonates with NEC at the clinical suspicion of the specific diseases. Sixteen gestational and postnatal age-matched neonates without sepsis/NEC served as controls. Serum metabolic profiles were assessed using liquid chromatography–quadrupole time-of-flight mass spectrometry. Metabolomic analysis revealed significant differences in the metabolic profile of neonates with LOS or NEC compared to controls. More specifically, a number of molecules possibly identified as phosphatidylcholines or lysophosphatidylcholines were found to be significantly reduced both in neonates with LOS and those with NEC compared to controls. Additionally, L-carnitine could efficiently discriminate NEC cases from controls. The results of the current study suggest that certain phospholipids and their derivatives could possibly be used as biomarkers for the early detection of LOS and NEC.

## 1. Introduction

Late-onset sepsis (LOS) and necrotizing enterocolitis (NEC) are major medical problems in neonates. LOS, most frequently defined as sepsis occurring more than or equal to 72 h of life, is a severe condition associating infection and systemic inflammatory response [1,2]. NEC is a potentially hazardous acute bowel disease, primarily characterized by inflammation of the intestine. Gut immaturity, enteral feeding, microbial dysbiosis, local ischemia, and/or reperfusion injury as well as inflammation are considered to play a significant role in the development of the disease [3]. Increased mortality and long-term morbidity in infants with LOS and NEC have been documented in several studies. Previous studies in very low birth weight infants reported a 36% risk for death in LOS due to Gram-negative microbes [4]. A lower mortality (6% in total and 10% in Gram-negative sepsis) was found recently in a randomized controlled trial of meropenem versus standard of care for treatment of neonatal LOS (NeoMero1) in preterm and term neonates [5]. NEC has been reported to be responsible for around 1 in 10 of all neonatal deaths [6]. In extremely low birth weight infants with surgical NEC, mortality was found to be as high as 50.9% [6]. Moreover, survivors from LOS and NEC are at increased risk for neurodevelopmental impairment [6,7]. 

Early diagnosis and timely prompt initiation of treatment are critical in both diseases. However, clinical signs and symptoms of LOS and NEC are non-specific and often overlapping. Moreover, the positivity of blood cultures (still considered the gold standard in neonatal sepsis) is around 50–60% [5,8], whereas the commonly used LOS/NEC biomarkers and radiological criteria (in NEC) have a relatively low diagnostic accuracy [3,9,10] making differential diagnosis difficult. 

Metabolomics is a new analytical method which detects metabolites of an organism in a biological sample [11,12]. These metabolites depict the state of the organism both in normal and pathological conditions and can be linked to specific biochemical pathways. In this context, metabolomics was found to play an increasingly important role not only in the investigation of the disease pathophysiology, but also in the discovery of diagnostic/prognostic biomarkers, opening new possibilities in the management of critically ill newborns [10,13]. Metabolomic studies in infants with sepsis and NEC are limited. To the best of our knowledge, urine metabolomics was evaluated in three studies on neonatal sepsis [14,15,16] and in three on NEC [17,18,19]. Feces have also been evaluated for the early detection of LOS [20] and NEC in preterm infants [21]. 

Partly due to the invasiveness of blood sampling, there is only one study of serum metabolomics in septic term neonates [22], and three studies in preterm infants with NEC in which serum was used for metabolomic analysis [23,24,25]. Alternatively, dried blood spot samples were also used [26]. Moreover, the metabolic profiles of both plasma and urine were recently evaluated in a case-control study involving preterm neonates with early-onset sepsis (<3rd day of life) [27].

Given the paucity of adequate evidence in the literature regarding metabolomics in neonates with LOS and NEC, and mainly on circulating metabolomics, we hypothesized that serum metabolic profiling could provide additional insights into the pathophysiology and diagnosis of LOS and NEC. Therefore, the aim of the current study was to evaluate metabolic changes related to the specific diseases and possibly detect novel early diagnostic biomarkers using non-targeted liquid chromatography–quadrupole time-of-flight mass spectrometry (LC–QTOF-MS).

## 2. Materials and Methods

### 2.1. Study Design and Population

This investigation was prospectively performed in a single, level III neonatal intensive care unit from September 2014 to August 2016. Actually, two cross-sectional, case-control studies were conducted: a study in neonates with LOS and a study in neonates with NEC. In both studies, cases were compared with the same control group. The latter consisted of subjects matched for gestational age (±one week) and day of life (±three days but always above 72 h of life) at the time of blood sampling, as gestational and post-natal age could affect blood metabolomic profiles in neonates [28].

Cases were selected from eligible preterm and term neonates who underwent evaluation for LOS or NEC, finally documented to suffer the disease(s). Diagnosis of LOS was based on criteria defined in 2010 by an Expert Meeting of the European Medicines Agency on Neonatal and Paediatric Sepsis [1]. Neonates with positive blood cultures for microbes or fungi were considered as having confirmed LOS, whereas those with clinical and laboratory evidence of sepsis and negative blood cultures were considered as having possible LOS. The modified Bell’s staging criteria (Appendix A) were used for the diagnosis of NEC [29].

Controls were identified among hospitalized neonates who were stable, less-severely ill infants (actually with no medical problems or recovering from minor ones) that, most importantly, had no clinical and laboratory evidence of sepsis or NEC upon enrollment.

Laboratory investigations for LOS/NEC cases included white blood cell counts and differentials, platelet counts, C-reactive protein, blood gas analysis (including lactate and glucose measurement), abdominal X-rays in NEC, as well as blood and urine cultures. Following the initial sepsis workup, cases were given treatment consistent with the protocols applied in our department (antibiotics, discontinuation of enteral feeding (always in NEC, and when clinically indicated in LOS), and, depending upon indications, supportive cardio-respiratory care, blood products transfusions, or surgery in NEC). Controls also underwent a sepsis workup upon study enrollment to rule out infection.

Neonates excluded a priori from enrolment were those born to mothers with clinical chorioamnionitis as well as those who had early-onset sepsis (onset in the first 72 h of life) or prior LOS/NEC, known congenital infections and anomalies (including gastro-intestinal ones), medium/severe hypoxic-ischemic encephalopathy, and inborn errors of metabolism. Refusal of parental consent, inadequate selection of blood samples, and physical absence of the researchers were also exclusion criteria. 

In all studied neonates, the demographic, perinatal, and clinical–laboratory data related to LOS and NEC were recorded.

An informed written consent was obtained from all parents/guardians before enrolling the neonates in the study. The study protocol was reviewed and approved by the Ethical Committee (Scientific Council) of the Hippokrateion General Hospital of Thessaloniki (7/4 September 2014). All work was conducted in accordance with the declaration of Helsinki of 1975 (https://www.wma.net/what-we-do/medical-ethics/declaration-of-helsinki/, accessed on 27 May 2022), revised in 2013.

### 2.2. Sample Collection and Preparation

#### 2.2.1. Sample Collection 

Blood samples were collected from cases at the clinical suspicion of LOS or NEC, and from controls upon enrollment. Blood samples were drawn through arterial lines or venipuncture of peripheral veins, simultaneously with routine tests. Next, samples were transferred into tubes with clot activator (Vacutest^®^, BD-Plymouth, Roborough, Plymouth, UK) and remained for 20 min at room temperature. Then, samples were centrifuged at 5000× *g* for 5 min and supernatants were stored at −80 °C until analysis. 

#### 2.2.2. Sample Preparation

Serum samples from LOS/NEC cases and control neonates were left to thaw at room temperature. All samples were vigorously vortex-mixed and an aliquot of 50 μL of each sample was transferred to an Eppendorf tube followed by the addition of 150 μL of cold acetonitrile, for protein precipitation. Afterwards, all samples were subjected to vortex-mixing for 1 min and centrifugation for 15 min at 10,000× *g* and 4 °C. The clear supernatants were transferred to LC–MS glass vials, placed on the auto-sampler at 10 °C, and subjected to LC–QTOF-MS analysis.

### 2.3. Liquid Chromatography–Quadrupole Time-of-Flight Mass Spectrometry (LC–QTOF-MS) Analysis

#### 2.3.1. Liquid Chromatography

All samples were analyzed using an Acquity UHPLC system (Waters, Elstree, UK) coupled to a Q-TOF Premier mass spectrometer (Waters MS Technologies, Manchester, UK). Liquid chromatography was performed on an HSS T3 Acquity column (2.1 mm × 100 mm, 1.7 μm) (Waters, Milford, MA, USA) attached to a VanGuard column (Waters, Milford, MA, USA) kept at 50 °C. Elution was achieved using a binary system consisting of mobile phase A, 10 mM ammonium formate + 0.1% formic acid, and mobile phase B, acetonitrile + 0.1% formic acid. The gradient applied was as follows: 0–1 min 95% A, 1–1.5 min 80% A, 1. 5–4.5 min 50% A, 4.5–9.5 min 5% A, 9.5–10.5 5% A, and 10.5–15.5 min 95% A. Flow was set at 0.35 mL/min and injection volume was set at 10 μL.

A pooled serum sample (quality control sample, QC), originated by mixing equal volume from all samples analyzed, was injected at the beginning of the analysis for system equilibration and thereafter every ten real samples throughout the batches as first proposed by our research group [30].

#### 2.3.2. Mass Spectrometry

MS was operated in positive electrospray mode (ESI+) at a scan range of *m/z* 70–900. Capillary voltage, source temperature, desolvation temperature, and desolvation gas flow were set at 3.5 kV, 150 °C, 350 °C, and 600 L/h, respectively. Cone voltage was set at 35 V and collision energy at 5 V. The Q-TOF was operated in V optics mode with a scan time of 0.5 s and an interscan delay of 0.1 s, and all data were collected in centroid mode. As a lock mass, 0.2 μg/mL of leucine-enkephalin (*m/z* 556.4528) solution was used.

QC samples were also analyzed in MS/MS positive V mode at the same scan range in order to obtain daughter ions of all features detected. Scan time was 0.5 s and low energy was set at 10 V with a ramp range of 20 to 30 V.

### 2.4. Data Handling and Statistical Analysis

All raw data were obtained using MassLynx 4.1 software (Waters, Wilmslow, Cheshire, UK). Conversion of .raw files to .mzxml was performed using the free available software MSConvert and then further processed with the XCMS online platform (The Scripps Research Institute, La Jolla, CA, USA; https://xcmsonline.scripps.edu/). XCMS online was used for feature detection, retention time correction, and alignment. Maximum mass accuracy was set at 10 ppm, peak width was between 14 and 40 s, signal to noise threshold was set at 10, and prefilter intensity was at 60,000 cps. The algorithm used was the centWave algorithm. For retention time correction, the obiwarp method was used with prodstep set at 1. Regarding the alignment, bandwidth was set at 5 s, minfrac at 0.5, and mzwid at 0.01. The extracted data matrix was additionally evaluated based on %RSD values of QC samples (%RSD ≤ 30%) and on the rule that a feature is presented at 80% of the samples of at least one group. The data matrix was normalized using the total area in MS Excel and quantile algorithm and log transformation in the MetaboAnalyst 5.0 online platform [31].

For multivariate statistical analysis, SIMCA P+ 13 (Umetrics, Malmö, Sweden) was used, where principal component analysis (PCA) and orthogonal partial least squares discriminant analysis (OPLS-DA) were performed in unit variance (UV) scaling for all comparisons. Resulting score plot models were evaluated based on the Q2Y, R2X, R2Y values, *p*-value of CV-ANOVA, and permutation plots. Visualization plots, as VIP (variable importance in projection), were additionally used for significant feature extraction.

Univariate statistical analysis was performed in both Microsoft Excel (Microsoft, Redmond, Washington, DC, USA) and GraphPad Prism 7.0 for Windows software (GraphPad Software, La Jolla, CA, USA; www.graphpad.com). FDR (false discovery rate) correction (Bonferroni–Dunn) was applied in all p-values. Threshold values were set at 0.05 for q-values, >1.0 for VIP, and >0.5 for p(corr).

Lipid identification was performed by matching accurate *m*/*z* measurements of statistically significant features to lipid species using available online databases (Metlin). For additional structural confirmation, tandem MS fragmentation pattern (from MS/MS experiments) was used; spectra were evaluated by comparison with those from online databases and literature. Most of the ions were identified at “Level 2” according to the identification level method of metabolites from untargeted analytical identification.

## 3. Results

### 3.1. Demographic and Clinical Data of the Study Population

We studied 15 neonates with LOS, 17 neonates with NEC, and 16 controls. The two disease groups and controls had comparable demographic/perinatal characteristics (Appendix A for LOS and NEC, respectively). Results of the clinical characteristics and the initial laboratory investigation for cases and controls are also shown in Appendix A. All but one sick neonate with sepsis were discharged home.

### 3.2. Untargeted Metabolomic Analysis

In total, 48 neonatal serum samples were analyzed randomly in a single analytical run applying a RPLC–TOF-MS method in positive ionization mode (ESI+). In order to assess the system’s stability, pooled serum quality control samples (QC) were analyzed every 10 real samples. In addition, five QC replicates were injected in the beginning of the analytical batch for column conditioning purposes in accordance to a protocol we described previously [32].

The obtained chromatographic data were further processed using the XCMS online platform, where 1091 features were extracted. Several filters were performed based on the CV value of QC samples (≤0.3) and the criterion of the features’ presence in at least 80% of a group. Furthermore, features eluted in the beginning of acquisition (<0.7 min) and at the end (>10.5 min) were excluded. The data matrix included 510 features and was used for further statistical analysis.

#### 3.2.1. Untargeted Metabolomics LOS

Univariate and multivariate statistical analyses were performed for the processed serum samples. The analytical performance proved to be satisfactory based on QC samples evaluation. OPLS-DA models showed a statistically significant difference between the sepsis and control groups (Figure 1). Characteristic values of the model were R2X = 0.889, R2Y = 0.796, Q2Y = 0.665, and CV-ANOVA = 0.005. Based on q-value, p(corr), and VIP values, eight metabolic variables were found to significantly contribute to the characterization of groups according to the adopted criteria (Table 1). Five out of the seven statistically significantly differentiated compounds were downregulated in the LOS group. Figure 2 illustrates the respective box plots of the most significant features based on the highest p(corr).

#### 3.2.2. Untargeted Metabolomics NEC

A combined approach of multivariate and univariate statistical analysis results was estimated to extract significant features for the pairwise group comparisons. The OPLS-DA model demonstrated a statistically significant differentiation between the NEC and control groups (R2X = 0.611, R2Y = 0.882, Q2Y = 0.484, CV-ANOVA = 0.002) (Figure 3). Nine features were found to be responsible for the clustering of the respective groups, following an upward trend except for *m/z* 496.3380 and *m/z* 522.3476. As can be clearly observed in the OPLS-DA score plot, the control group presented a smaller deviation compared to the disease group, where the metabolome was affected by a variety of factors. Table 2 summarizes the statistically significant features with their respective q-value, p(corr), and VIP values, while Figure 4 depicts the respective box plots of the most significant features based on the highest p(corr).

#### 3.2.3. Metabolites Identification

Lipid (tentative) identification was performed utilizing high-resolution mass spectra in MS and MS mode and aimed at recognizing head groups and lipid species. (Lyso)PCs can be identified in positive ionization mode as sodium and potassium adducts as well as in their protonated cationic form. Among the statistically significant features, lipids comprising the class of phosphatidylcholine(s) (PC)s were mainly identified covering different polarities. Identification was performed by a combination of either automatic and/Tor manual data manipulation and library searching. For all identified PCs, a characteristic head-group fragment *m/z* 184.0739 (C5H15NO4P+) was observed. In Figure 5, an example of a manually annotated PC is presented.

## 4. Discussion

In this prospective study, we investigated the role of metabolomic analysis using non-targeted LC–QTOF-MS in identifying possible serum biomarkers in neonates with LOS or NEC. To our knowledge, this is the first study evaluating alterations of serum phospholipids in neonatal inflammatory conditions such as LOS and NEC. Analysis demonstrated significant differences in the metabolic profiles of neonates with LOS or NEC when compared with the corresponding controls. The significant metabolites identified both in LOS and NEC belonged to subcategories and derivatives of phospholipids. Additionally, L-carnitine could efficiently discriminate NEC cases from controls.

### 4.1. Neonates with LOS

We observed a significant difference in seven metabolites in neonates with LOS. Five of them were identified as glycerolphospholipids (phospholipids), while two were unknown. The above phospholipids were possibly identified as PCs or lysophosphatidylcholines (LysoPCs), and were found to be reduced in septic neonates compared to the controls.

Phospholipids comprise a class of complex lipids containing a phosphate group and two fatty acid derivatives joined by an alcohol residue. Simple molecules, such as choline, ethanolamine, or serine, can further modify the phosphate group forming the corresponding derivatives. Glycerophospholipids (PC, phosphatidylethanolamine (PE), phosphatidylserine (PS), phosphatidylinositol, PG, phosphatidylglucoside, cardiolipin) are major phospholipids characteristically containing glycerol [33].

PC is the prominent component of cell membranes (40–50% of all cell phospholipids) [33]. In mammals, PC is produced in the liver via the CDP-choline pathway (also called the Kennedy pathway) and following the conversion of PE to PC (phosphatidylethanolamine N-methyltransferase (PEMT) pathway) [34]. LysoPCs are derived from the hydrolysis of phospholipids mainly by the enzyme phospholipase A2. For instance, LysoPC and fatty acids (arachidonic acid) are formed following PC hydrolysis [35]. In addition to their role in membrane formation, phospholipids (including PC) and their products are important intracellular mediators and enzyme activators [33].

Several previous studies have reported a severely disturbed lipid homeostasis in sepsis. In line with our results, in an experimental model of septic shock, a marked decrease in plasma PCs (and LysoPCs) was observed (on the 1st and mainly on the 7th day of sepsis) in the animals who died compared to those who survived, associating low levels of PCs with high mortality [36]. In contrast, Schmerler et al. reported increased levels of the metabolic markers C10:1 and PC32:0 in the plasma of septic patients that enabled the discrimination of septic from non-septic patients [37].

Regarding LysoPCs, our findings are in parallel with the results of previous studies in septic patients reporting decreased plasma LysoPC levels [38,39] that were correlated with increased mortality [38,40,41]. In addition, a greater reduction of LysoPCs was noted in patients with bacterial infection when compared to those with non-bacterial infection [38,39]. Nevertheless, no difference was found between patients suffering from severe sepsis versus septic shock [41]. In septic patients who died, a significant reduction of the LysoPC/PC ratio was also reported. A recent study, using an experimental model of multibacterial peritoneal sepsis in rats, demonstrated decreased levels of LysoPC, albumin, and activity of enzymes entering the metabolism of LysoPC, while PC, lysophosphatic acid, and LysoPC acyltransferase levels were increased. Decreased circulating LysoPCs might hamper their anti-inflammatory effects and, conceivably, may induce an excessive immune response [36].

It is worth noting that of all LysoPCs identified in the present study, LysoPC (16:0/0:0) seems to be the most promising as a sepsis biomarker [41,42].

### 4.2. Neonates with NEC

In neonates with NEC, nine different metabolites were isolated, of which three were possibly identified as glycerolphospholipids and one as L-carnitine. Compared to controls, glycerophospholipids PC (16:0/0:0) or LysoPC (16:0/0:0) as well as PC (18:1/0:0) or LysoPC (18:1/0:0) were found to be significantly decreased, and PC (20:4/0:0) significantly increased in the cases.

PCs (mainly PC 16:0/18:1 and PC 16:0/18:2) and LysoPCs (LysoPC 16:0/0:0) are the dominant phospholipids in the mucus of the intestinal barrier, providing protection to the epithelial cells of the gut by forming a hydrophobic mucus layer [43]. In accordance with our results, studies in adult patients with inactive ulcerative colitis showed significantly decreased PC and LysoPC in rectal mucus (but not in PE), thus explaining the increased susceptibility to luminal contents [44]. Different mechanisms have been proposed to explain the anti-inflammatory action of the exogenous PC [43]. In this context, the decrease of certain PCs and LysoPCs in our NEC cases may indicate an involvement of these anti-inflammatory agents in the gut host defense mechanisms. On the contrary, the increase in PC (20:4/0:0) may be associated with the destruction of gut cells following NEC. Still, the fact that PC may also be synthesized by certain bacteria [45] does not preclude the bacterial origin of the circulating PC and LysoPC in neonates with NEC.

L-carnitine, a non-protein amino acid [46], discriminated neonates with NEC from those without the disease. L-carnitine facilitates beta-oxidation of the long and very long chain fatty acids for energy production. Moreover, by increasing antioxidant enzyme activities, it suppresses the production of free oxygen radicals, reducing oxidative stress [47]. In experimental NEC, the exogenous administration of L-carnitine was reported to ameliorate the histological evidence of intestinal injury and decrease lipid peroxidation [48]. Thus, increased L-carnitine in our neonates with NEC probably reflects increased cell demands for energy production from fatty acids, as well as a compensatory mechanism mitigating the toxic effect of free oxygen radicals.

This study adds new information to the limited data regarding serum/plasma metabolomics in neonates with LOS or NEC. Although blood sampling is a more interventional approach compared to urine, blood remains the most studied specimen in particular to monitor serious or life-threatening situations. Blood serum is expected to better reflect pathophysiological and metabolic changes in systematic illness, such as sepsis and NEC. On the other hand, an obvious restriction of this study is the fact that it was conducted in a single center and, consequently, the number of participating neonates was limited. Based on previous metabolic studies in LOS and NEC from our research group [16,18], the sample size of the present investigation was adequate to document differences in the metabolic profiles. However, we could not evaluate the impact on the metabolic profiles of other important variables, such as the type of microbes, cases of confirmed versus possible sepsis/NEC, feeding mode, and mortality.

## 5. Conclusions

At the onset of clinical presentation, neonates with LOS and NEC develop different serum metabolomic profile compared to controls, allowing their clear separation with the use of LC–QTOF-MS. Significant difference in metabolites include certain phospholipids and their derivatives such as PCs/LysoPCs, which in turn could potentially be used as surrogate biomarkers for the early detection of LOS and NEC.

## Figures and Tables

**Figure 1 jcm-11-05270-f001:**
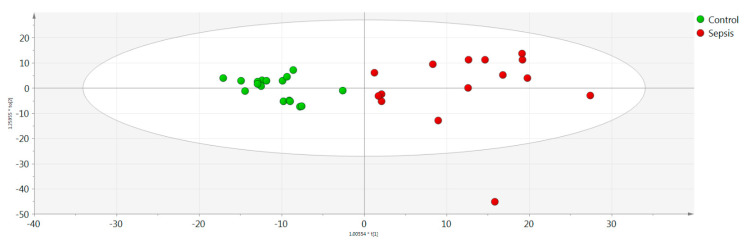
Orthogonal partial least squares discriminant analysis (OPLS-DA) score plot model between the control and late-onset (LOS) groups. Y and X axes represent [t2] and [t1], respectively.

**Figure 2 jcm-11-05270-f002:**
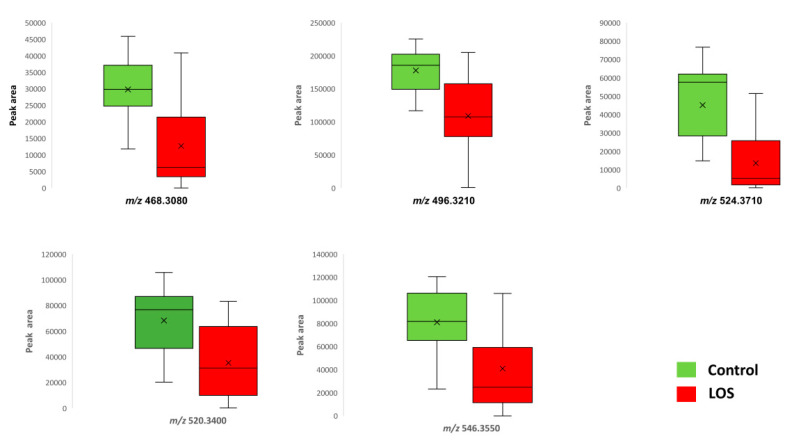
Box plots of significant metabolites in the control and late-onset (LOS) groups.

**Figure 3 jcm-11-05270-f003:**
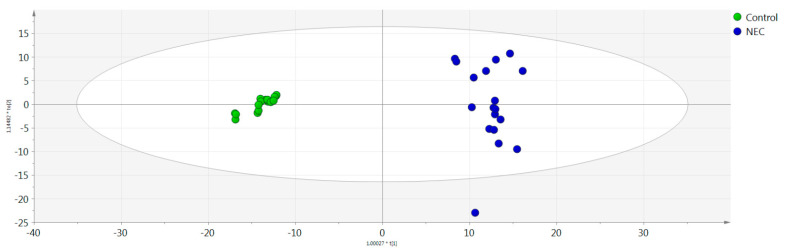
Orthogonal partial least squares discriminant analysis (OPLS-DA) score plots of the control and necrotizing enterocolitis (NEC) groups. Y and X axes represent [t2] and [t1], respectively.

**Figure 4 jcm-11-05270-f004:**
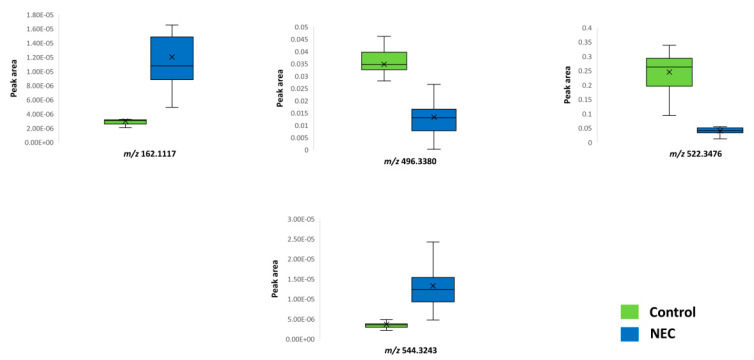
Box plots of significant metabolites in the control and necrotizing enterocolitis (NEC) groups.

**Figure 5 jcm-11-05270-f005:**
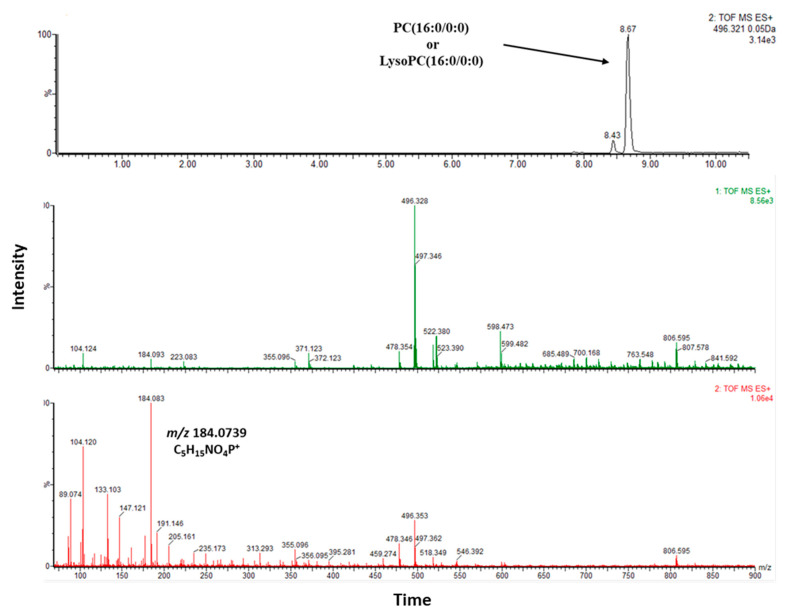
Chromatographic peak of phosphatidylcholine (16:0/0:0) or lysophosphatidylcholine (16:0/0:0) along with MS and MS/MS spectra.

**Table 1 jcm-11-05270-t001:** Statistically significant features with their respective q-value (Bonferroni), p(corr), and VIP values.

*m/z*_RT	VIP	p(corr)	q-Value	Possible Identity
468.308_7.5	3.5	−0.71	3.7 × 10^−4^	PC (14:0/0:0) or LysoPC (14:0/0:0)
496.321_8.6	6.8	−0.66	1.2 × 10^−3^	PC (16:0/0:0) or LysoPC (16:0/0:0)
520.340_8.0	4.3	−0.52	3.7 × 10^−3^	PC (18:2/0:0) or LysoPC (18:2/0:0)
524.371_9.5	4.4	−0.67	3.6 × 10^−4^	PC (18:0/0:0) or LysoPC (18:0/0:0)
546.355_8.5	4.6	−0.55	5.4 × 10^−3^	PC (20:3/0:0) or LysoPC (20:3/0:0)
594.378_9.2	1.5	0.64	2.7 × 10^−2^	Unknown
638.939_9.0	1.4	0.55	5.4 × 10^−2^	Unknown

**Table 2 jcm-11-05270-t002:** Statistically significant features with their respective q-value (Bonferroni), p(corr), and VIP values.

*m/z*_RT	VIP	q-Value	p(corr)	Possible Identity
120.0808_2.1	1.4	3.3 × 10^−5^	0.73	Unknown
146.0593_2.9	1.34	2.1 × 10^−4^	0.71	Unknown
162.1117_0.7	1.33	3.1 × 10^−4^	0.71	L-carnitine
188.0693_2.9	1.36	1.3 × 10^−4^	0.72	Unknown
225.1968_6.3	1.36	9.2 × 10^−5^	0.73	Unknown
272.2582_6.7	1.3	4.0 × 10^−4^	0.74	Unknown
496.3380_8.4	1.28	1.4 × 10^−3^	−0.78	PC (16:0/0:0) or LysoPC (16:0/0:0)
522.3476_9.0	1.52	1.4 × 10^−3^	−0.87	PC (18:1/0:0) or LysoPC (18:1/0:0)
544.3243_8.2	1.39	4.8 × 10^−3^	0.73	PC(20:4/0:0)

## Data Availability

All relevant data are contained within this manuscript.

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
