# Peer review of "A Prospective, Case-Control Study of Serum Metabolomics in Neonates with Late-Onset Sepsis and Necrotizing Enterocolitis"

_jcm, 2022, doi:10.3390/jcm11185270_

Round 1

Reviewer 1 Report

I did not understand why the same text was repeated in the Introduction part (35-79 lines) and the Materials and methods part (81-125 lines) and 126-170 lines. Maybe it is a mistake in the submission? And there are no Materials and methods at all described- which level hospital, which patients and how were selected, how informed content was obtained, how controls were selected, etc. Which machine was used for the detection of metabolomics? How statistics was made?

I did not find an explanation of the criteria of proven LOS and possible LOS. In my opinion, it has to be included n the text.

Reviewer 2 Report

Reviewer’s comments:

Title: A prospective case-control study of serum metabolomics in neonates with late-onset sepsis and necrotizing enterocolitis.

Overview: This is an interesting case control study evaluating differences in metabolomics between 3 groups of neonates- those with NEC, late-onset sepsis defined by , and control infants without NEC or late-onset sepsis. However, there are no methods mentioned, instead there is repetition of introduction under methods subheading. The authors observed decreased phosphatidyl choline, lysophosphatidyl choline among infants with LOS but not in control infants.  They also observed decrease in L-carnitine and other abnormalities in metabolomics among infants with NEC but not in control infants.

Abstract:

Are “Phophatodyl cholines” and “lysophosphatodylcholines” the same as phosphatidyl cholines and lysophosphatidyl cholines? Please correct them.

Introduction

Line 46-47: “NEC has been reported to be responsible for around 1 in 10 of all neonatal deaths”- Is this true? please add reference.

Line 57: “…. Detects metabolites of an organism in a biological sample”. Can the authors check if it should read as “or a biological sample”

Methods

Lines 81- 170 are just a repetition of the introduction. There are no methods mentioned in this section.

Results

Lines 192- 206 should be under methods.

Discussion

Lines 257-264: well-written.

Tables

Table S1: How do the authors define “confirmed LOS”, and “possible LOS”

Table S2: How do the authors define “conformed NEC” and “suspected NEC”.

Figures: What are the units for the x- and y- axes?

Author Response

Abstract:

Are “Phophatodyl cholines” and “lysophosphatodylcholines” the same as phosphatidyl cholines and lysophosphatidyl cholines? Please correct them.

RE: The corrections were done.

Introduction

Line 46-47: “NEC has been reported to be responsible for around 1 in 10 of all neonatal deaths”- Is this true? please add reference.

RE: This sentence has the same reference as the following one regarding extremely low birth weight infants. In any case, it is now clearly shown that both statements have the same bibliography source.

Line 57: “…. Detects metabolites of an organism in a biological sample”. Can the authors check if it should read as “or a biological sample”

RE: We believe that the sentence “Detects metabolites of an organism in a biological sample” is more appropriate.

Methods

Lines 81- 170 are just a repetition of the introduction. There are no methods mentioned in this section.

RE:  As also mentioned in our reply to the other reviewer, this was an error of the system creating the article’s journal format. A separate “Materials and Methods” paragraph was present already at our initial submission.

Results

Lines 192- 206 should be under methods.

 RE: We think that the text should remain in its present format, as the specific paragraph describes the results of the set criteria previously shown in data handling.

Discussion

Lines 257-264: well-written.

Tables

Table S1: How do the authors define “confirmed LOS”, and “possible LOS”

Table S2: How do the authors define “conformed NEC” and “suspected NEC”.

RE: It was already written in the manuscript that “Neonates with positive blood cultures for microbes or fungi were considered as having confirmed LOS, whereas those with clinical and laboratory evidence of sepsis and negative blood cultures as having possible LOS. The modified Bell's staging criteria (Supplementary material) were used for the diagnosis of NEC [29].

Figures: What are the units for the x- and y- axes?

RE: The titles of the y and x axes in figures 1 and 3 are respectively [t2] and [t1]. This is now written in the legends of the specific figures. The titles of the axes in Figures 2, 4 and 5 are already shown.